# Modification and Functionalization of Fibers Formed by Electrospinning: A Review

**DOI:** 10.3390/membranes12090861

**Published:** 2022-09-06

**Authors:** Gabriela B. Medeiros, Felipe de A. Lima, Daniela S. de Almeida, Vádila G. Guerra, Mônica L. Aguiar

**Affiliations:** 1Departamento de Engenharia Química, Federal University of São Carlos, Rodovia Washington Luiz, km 235-SP 310, São Carlos 13565-905, SP, Brazil; 2Departamento de Engenharia Ambiental, Federal University of Technology-Paraná, Avenida dos Pioneiros, 3131, Londrina 86030-370, PR, Brazil

**Keywords:** electrospinning, nanotechnology, fiber modifications

## Abstract

The development of new materials with specific functionalities for certain applications has been increasing with the advent of nanotechnology. A technique widely used for this purpose is electrospinning, because control of several parameters involved in the process can yield nanoscale fibers. In addition to the production of innovative and small-scale materials, through structural, chemical, physical, and biological modifications in the fibers produced in electrospinning, it is possible to obtain specific properties for a given application. Thus, the produced fibers can serve different purposes, such as in the areas of sensors, catalysis, and environmental and medical fields. Given this context, this article presents a review of the electrospinning technique, addressing the parameters that influence the properties of the fibers formed and some techniques used to modify them as specific treatments that can be conducted during or after electrospinning. In situ addition of nanoparticles, changes in the configuration of the metallic collector, use of alternating current, electret fibers, core/shell method, coating, electrospray-coating, plasma, reinforcing composite materials, and thermal treatments are some of the examples addressed in this work. Therefore, this work contributes to a better comprehension of some of the techniques mentioned in the literature so far.

## 1. Introduction

Nanotechnology has facilitated the creation of materials that have provided significant advances in areas such as energy, electronics, and medicine. The prospects are promising: nanotechnology may enable an improvement in life quality in diverse sectors. Quina [1] mentioned the potential of nanomaterials to increase the efficiency of processes and, consequently, make more efficient the use of raw materials, with lower energy consumption and less waste generation. In this sense, nanomaterials have been applied in several areas, such as the catalytic area, to increase the efficiency and selectivity of industrial processes [2]; in the therapeutic field, to obtain better ways of assimilation of medicines by the human body [3,4,5]; in the cosmetic area, with the development of better product delivery and absorption systems [6]; and other areas, including the manufacture of supercapacitors and batteries [7,8].

Nanofibers and nanoparticles are materials with a thickness or diameter below the micrometer (sub-micrometric dimensions). Nanofibers and nanoparticles with the same diameter have different surface areas. Considering the same diameter, generally, nanoparticles exhibit a larger surface area if compared to nanofibers. However, when there is a need to reuse nanoparticles, this poses a difficulty when separating them from the dispersion medium. Therefore, a combination of the properties of nanoparticles and nanofibers could be an interesting strategy [9]. Currently, several technologies have been developed to manufacture nanofibers, such as the stretching method, “island” spinning, melt spinning, solution spinning, emulsion spinning, and electrospinning [10,11].

The electrospinning technique is widely used in the production of nanofibers since the produced fibers have a uniform appearance, in addition to allowing the use of various polymeric solutions, with a very wide range of applications, like in the areas of separation processes; sensors; catalysis; environmental; and medical [12,13,14]. Thus, with this review, it is possible to better understand the process of fiber formation by the electrospinning technique and also to get in touch with the main fiber modification techniques to functionalize them for a variety of applications. 

## 2. The Electrospinning Process

One of the most versatile methods of producing nanofibers is electrospinning. It is an easy-to-operate, low-cost technique that involves the application of high-voltage between the tip of a needle and a metallic collector (Figure 1). The polymer solution or melt material contained in the syringe is injected, with its flow controlled by a pump. When the drop of solution leaves the tip of the needle, the electric field causes deformation and stretching of the drop, forming a Taylor cone. The elongation of the cone forms a viscoelastic jet that is pulled away from the tip by the electric field and gravity altogether. The tensile force is generated by the interaction of the electric charge carried by a jet with the applied electric field. This force is transmitted by the viscous flow and viscoelasticity of the liquid in the jet and also by surface tension. On the way to the collector, the evaporation of the solvent and solidification of the fibers in the collector occurs [12,15].

The formed fibers present morphologies and diameters depending on the solution, process, and environmental parameters used in electrospinning. The principal solution parameters are viscosity (related to polymer concentration and molar mass), conductivity, surface tension, dipole moment, and dielectric constant of the solvent. The process parameters are flow rate, applied electric field, and the distance between needle and collector (working distance). Ambient conditions that influence the process are relative humidity, temperature, and air velocity in the electrospinning chamber [12,16,17]. Based on these factors, it is essential to control the parameters during the process to analyze how each one will influence the properties of the formed fibers.

### 2.1. Influence of Solution, Process and Environmental Parameters

The principal solution parameters that affect the diameter of the resulting fibers and the formation of beads (small drops solidified along the length of the fiber, as illustrated in Figure 2) are polymer solution concentration, viscosity, conductivity, superficial tension, dipole moment, the dielectric constant, and volatility of the solvent. These parameters mainly influence the extent of jet elongation that occurs from the needle tip to the collector [16,18,19].

One of the determining parameters in electrospinning is viscosity. It is directly related to the concentration of the solution and the molar mass of the polymer used. At low polymer concentrations (low viscosity values), the electrospun jet breaks up on its way to the collector, due to surface tension. Thus, no fibers are formed. In this case, the formation of polymer droplets (electrospraying) or fibers with beads occurs due to viscoelastic forces that would not dampen the breakage mechanism. As the concentration of the polymeric solution increases (higher viscosity values), the probability of fiber formation increases, as the viscoelastic forces are more pronounced. However, the diameter of the electrospun fibers also increases with increasing solution viscosity. Another disadvantage of using very viscous solutions is that they can cause the needle to clog during the electrospinning process [18,20].

Surface tension is the force exerted on the surface plane per unit length. The surface tension of the solution is a parameter that influences the kinetics of Taylor cone formation and electrospinning process start. The surface tension of the drop must be overcome by electrostatic forces for the jet and Taylor cone formations. Thus, it was observed that smooth fibers are formed without the presence of beads at reduced values of surface tension of the solution [21]. In solutions that have high electrical conductivity, there will be mobility of ions in the solution. When an external electric field is applied, the ionic charges are oriented, occasioning the deformation of the solution drop. After the formation of the Taylor cone, the process of ejection and elongation of the jet contributes to the decrease in the final diameter of the formed fiber. Higher conductivity values can be obtained by adding salt [16,21]. However, the addition of salt can affect the viscosity of the solution, in addition to its presence in the dry nanofibers, which can prevent its use in some applications, making a post-treatment necessary for removal. Promising results were also found with the addition of a co-solvent to the solution to increase the conductivity, such as alcohol [16]. In some cases, the addition of salt is necessary for the complete dissolution of the polymer [22].

Besides the properties of the polymer used in electrospinning, the choice of solvent used in the preparation will also influence the morphology and diameter of the fibers produced. Solvents that present greater efficiency in the electrospinning process have high dipole moments, such as N, N-dimethylformamide (DMF), ethyl-methyl-ketone (MEK), and ethyl acetate [23]. When the first drop of solution comes out of the needle tip, the solvent starts to evaporate. Thus, its volatility is one of the parameters that influences the electrospinning process. More volatile solvents facilitate the collection of dry fibers. However, they can cause needle clogging as the solvent evaporates and the polymer solidifies at the needle tip. Solvents with lower volatility allow a longer stretching process until complete solidification, therefore, they produce finer fibers [24,25]. The addition of another solvent can alter the concentration of a solution and improve the electrospinning process and the nanofiber’s quality [26].

One of the main influencing factors in the process is the flow rate of the solution [12,16,19]. Studies show that at low flow rate values, fibers with smaller diameters are produced. In this case, there is less bead formation. However, a minimum rate is required so that there is no lack of a solution to be injected and not produce defects in the fibers. A greater flow produces a larger drop in the capillary, which results in a faster trajectory of the solution to the collector, and with that, the evaporation of the solvent is incomplete and the formation of a greater bead occurs, in addition to producing solvent-wet fibers [20,21].

The applied electrical voltage is another parameter that has been analyzed. The voltage must be sufficient to overcome the surface tension of the droplet and form a Taylor cone. Depending on the type of polymer solution, there is an optimal range of voltage in which fiber formation occurs. At voltages outside the optimal range, greater bead formation occurs or the polymer jet is inhibited. Smaller final fiber diameters are obtained at higher electric fields since the increase in the electric field causes the jet to elongate [20].

The distance between the needle and the collector, called the working distance, is another parameter that can be optimized in search of smaller diameters and adequate morphology. There must be a minimum distance so that all of the solvent is evaporated during the journey to the collector and there is no formation of beads or even flattened fibers. More uniform, cylindrical fibers are found at greater distances [16,20].

A setup option parameter that significantly influences the morphology of the fibers is the type of collector, as well as its geometry. As an electric field is applied between the needle and collector during electrospinning, the collector must be made of an electrically conductive material such as aluminum or copper. There are two geometries for the collector, a metallic plate or a rotating metallic collector. When using a rotating collector, the morphology and diameter of the fibers will be influenced by the rotation speed of the collector [16].

The local conditions in which electrospinning is conducted, such as relative humidity and temperature, are the environmental parameters that influence the fibers formed. High temperatures cause a decrease in the viscosity of the polymeric solution and, as a result, fibers with smaller diameters can be obtained [16,27]. Thenmozhi et al. [21] presented a study of the influence of collector temperature on fiber morphology. The authors show that ambient temperature values close to the boiling temperature of the used solvent increase the number of pores on the surface of the fibers. This is because the residual solvent evaporates. At temperatures higher than boiling, the evaporation of the solvent is accelerated. Thus, the number and size of pores in the fibers increase.

When evaluating the effect of relative humidity, the results are different in different studies. Moisture/fiber diameter ratios will depend on the polymers used in the process. For example, Yan, and Gevelber [28] observed that for poly(ethylene oxide) (PEO), the average fiber diameter decreases with increasing moisture. Thenmozhi et al. [21] showed the relationship between humidity and temperature: as the temperature increases, the humidity decreases, since the evaporation of the solvent is faster. Due to this fact, high humidity leads to the formation of thicker fibers.

### 2.2. Fiber Modifications during the Electrospinning Process

#### 2.2.1. Use of Flat Plate Collectors and Rotating Cylinders

The process variables, the solution, and environmental parameters can be analyzed to find optimal experimental conditions for the production of nanofibers for a specific application. Thus, it is possible to combine the polymer used to the process conditions to obtain fibers with adequate properties depending on the type of application. Figure 3 shows a summary of the main electrospinning parameters and their effects on fiber diameter. An important aspect of the electrospinning process is the type of collector used. In this process, a collector serves as a conductive substrate where the nanofibers are collected. Generally, aluminum flat plate is used as a collector (Figure 4a), but due to the difficulty in transferring the collected fibers and the need for aligned fibers for various applications, other collectors, such as rotating rods and drums (Figure 4b), started to be investigated [29].

Due to the bending instabilities of the jets, the fibers obtained with electrospinning are often deposited on the surface of the collector randomly. Several approaches try to control the jet motion by the electric field distribution and demonstrate a tendency for the collected fibers to be arranged in a uniaxial way [30]. However, this is a very difficult goal to achieve for electrospun nanofibers because the polymer jet trajectory has a very complicated three-dimensional shape caused by bending instabilities rather than being in a straight line [13]. In the case of plates such as the collector, the jet is moved laterally and forms a series of coils, whose envelope has the shape of a cone opening towards the counter electrode, leading to entirely random and unaligned depositions [12].

The use of cylindrical and rotating collectors, rotating at a very high speed, can orient the fibers circumferentially. However, fiber alignments were only achieved to a certain extent and small diameter fibers remained randomly oriented [13]. Another way is to use two strips of metal separated by a void of several centimeters, causing changes in the electrostatic forces acting, which causes the charged fibers to be stretched to align themselves perpendicularly [30]. Efforts are being made in various research groups around the world, but so far, publications related to aligned nanofibers are very limited.

The main difference between the types of collectors is in the alignment of the fibers. Studies show that more aligned fibers achieve improved tensile properties [31]. This improvement in mechanical properties, especially in tensile strength, can be explained by the fact that its orientation is the same as the applied force, which makes it difficult to break the fibers. The random fibers will only have part of them in the direction of the applied force [32]. Chen et al. [31] observed a Young’s modulus for aligned fibers 8.5 times greater than for random fibers. Merighi et al. [32] also observed an improvement in the modulus of elasticity, about 5 times greater. The aforementioned studies show an improvement in the mechanical properties of the fibers produced by rotating collectors (aligned fibers).

#### 2.2.2. Core/Shell Fibers

Coaxial electrospinning or co-electrospinning (core/shell) is a process that appeared to allow greater functionalization of nanofibers through the introduction of functional molecules or objects, as illustrated in Figure 5 [12]. In certain fields, such as biosensor technology, tissue engineering, drug delivery, and nanoelectronics, including active agents or different functional properties on the surface of the nanofibers (shell), while maintaining the intrinsic properties of the nanofibers (core), would allow greater applicability, as in biotechnology, drug delivery, and nanofluidics [12,33,34]. Another point is to allow low molecular weight materials to be transformed into fiber, using another polymer together to improve mechanical properties [12].

In general, two methods are most used for the production of core/shell nanofibers: emulsion and coaxial electrospinning. In emulsion electrospinning, an emulsion of two immiscible polymers is fed into a syringe with a capillary nozzle, while in coaxial electrospinning, the core and shell solutions are fed separately into a coaxial nozzle. Between those two, coaxial electrospinning is generally considered to be one of the most versatile methods for manufacturing these types of nanofibers [12].

It is important to note that an emulsion electrospinning can be conducted using immiscible polymers. However, it is not mandatory for two immiscible polymers leading to the formation of a core-shell or multi core-shell nanofibrous structure. Indeed, the polymer pair can form a homogeneous solution, and in such a case, the resulting nanofiber is a blend of the homopolymers. In general, when two polymers are blended, they tend to form two phases due to adverse thermodynamic effects, limiting the number of fully miscible polymer pairs [35,36]. However, several studies are making blends of two polymers thermodynamically favorable, and the number of known miscible pairs has increased by an order of magnitude or more [36]. The creation of blend nanofibers or core-shell ones may also derive from the particular polymer pair, as demonstrated by the electrospinning of nitrile rubber (NBR) with polycaprolactone (PCL) or Nomex: in the first case, blend nanofibers are obtained [37], while in the other case, nanofibers reminiscent of core-shell nanofibers are obtained [38].

In contrast to coaxial electrospinning, the behavior of nanofibers containing two polymers with high phase separation tendance produced by solvent evaporation during a single spinneret electrospinning is expected to be more complex [39]. Aluigi et al. [35] evaluated the effects of the degree of miscibility on the morphology and supermolecular organization of keratin and polyamine-6 cast films. Thew showed that with low miscibility, the surfaces tend to become more granular and the segregation of phases between polymers is evidenced. The authors were also able to relate the morphology of nanofibers with process parameters. Quan et al. [40] were able to establish the model of the core-sheath structure-forming mechanism. They used the phase-changing material dodecanol and the polymer polyvinyl alcohol as the oil phase and water phase, respectively, to prepare the oil-in-water (O/W) core-sheath structured nanofibers. Other studies evaluated the impact of the fibers produced by this technique in the performance of some process. For example. Park et al. [41] obtained a unique multicore-shell structure of electrospun composite fibers of polyimide and polyvinylidene fluoride, and there was a synergy between the thermal stability of polyamide with the polyvinylidene fluoride shell as electrolyte compatible polymers that allowed a better thermal performance and long-term stability of Li-ion battery. Tipduangta et al. [39] applied blends of polyvinylpyrrolidone and hydroxypropyl methylcellulose acetate succinate in controlled drug delivery (paracetamol). The results demonstrated an improvement in the adjustment of the rate of drug release, which can be adopted to deliver two drugs in a single formulation that has different targeted release areas desired in the intestine.

In coaxial electrospinning, two concentrically aligned nozzles are used and a coaxial jet of two different liquids flows through the outer and inner capillaries simultaneously. The same voltage is applied to both nozzles and deforms the droplet. In an ideal case, a core/shell nanofiber is created and consolidated during solvent evaporation and stretching. The two-component feed ratio affects the uniformity and stability of the fiber [12]. 

A jet is generated at the tip of the deformed droplet, and in an ideal case, a core-shell nanofiber is created. Upon closer inspection, it becomes clear that coaxial electrospinning involves a set of intricate physical processes, which have only been elucidated experimentally to some extent and require mathematical modeling [42,43]. Other parameters such as core-shell capillary size, applied electric field, volume feed rate, immiscibility of core/shell liquids and their viscosity and conductivity also play a crucial role in determining the uniform formation of core/shell jets and the morphology of the nanofibers produced in this electrospinning method [42].

Although coaxial electrospinning requires more complicated configuration adjustments than emulsion electrospinning, it still has significant advantages due to certain shortcomings of the latter. Emulsion electrospinning is restricted to immiscible polymer electrospinning, while separate channels in coaxial electrospinning make it possible to manufacture core-shell fibers even for two miscible polymers. Furthermore, in some cases of emulsion electrospinning, instead of a continuous core, separate bubbles of core embedded in the shell polymer will be formed, which is not desirable.

Many studies have used coaxial electrospinning to produce special nanofibers. Weerasinghe et al. [44] carried out a project to manufacture fully organic, conductive, and biodegradable core/shell fibers. For this, in situ polymerization of aniline was carried out on the surface of electrospun poly(ε-caprolactone) nanofibers. Electrical resistance changed almost instantly with tension for several stretch and recovery cycles. This rapid and sensitive response to mechanical loading and unloading holds promise for validating the possibility of using lead wires as strain sensors to monitor human movement; increasing the number of yarn layers to three resulted in a threefold reduction in strength.

Ji, Tiwari, and Kim [45] reported the synthesis of core/shell nanofibers of graphene oxide (GO) encased in zinc oxide (ZnO) by single coaxial electrospinning and subsequent annealing for improved photocatalytic performance and stability. The heterostructured catalyst consisted of ZnO forming a closed central part while GO was positioned on the surface, serving as a protective shell. The photocatalytic activity of these fibers was increased in comparison to those of simple ZnO. Furthermore, the incorporation of GO into the ZnO nanofiber in a core-shell format significantly suppressed photo corrosion.

Maleknia et al. [46] evaluated the production of core/shell fibers with polyurethane (PU) and chitosan (Cs)/polyethylene oxide (PEO). The PU/C nanofibers were spun without any structural imperfections under the optimized processing conditions, without significant variations in crystallinity. It has been hypothesized that core/shell nanofibers of PU/Cs could be used as a potential platform to deliver bioactive molecules in a sustained manner in tissue engineering.

Nie et al. [47] prepared nanofibers by coaxial electrospinning with poly(ethylene oxide) (PEO) as core and poly(acrylic acid) (PAA) as cladding. PEO and PAA can form polymer complexes based on hydrogen bonds. In order to avoid the formation of strong hydrogen-bonding complexes in the nozzle and blocking the rotation process, a polar aprotic solvent, N,N-dimethylformamide (DMF), was selected to dissolve PEO and PAA, respectively. It was evidenced that the PAA shell of nanofibers can be crosslinked by ethylene glycol (EG) under heat treatment to increase its stability and extend its application potential in aqueous media.

Rahmani, Arefazar, and Latifi [48] evaluated coaxial electrospinning for the production of polymethylmethacrylate (PMMA) and polystyrene (PS) core-shell fibers. To evaluate the influence of the solvent on the final fiber morphology, four types of organic solvents were used in the shell solution, while the core solvent was not changed. Experimental observations revealed that the solvent properties of the core and shell were involved in the final fiber morphology. To explain this involvement, a theoretical model based on the principle of conservation of momentum was developed and applied to describe the dependence of core and shell diameters on their solvent combinations. The Bagley solubility plot was used for core and shell solvent selection, with the introduction of a factor (β) associated with the selection of the most likely successful core and shell solvents. By the proposed mathematical model, the surface tensions, dielectric properties, and power-law characteristics of the core and shell solutions are the significant solvent and solution parameters that define the diameters of the core and shell fibers. Other morphological aspects of the fibers depend on parameters such as surface tension, viscosity, voltage, and solvent evaporation rate.

Forward, Flores, and Rutledge [49] proposed a new methodology to increase process productivity through the development of coaxial jets directly from droplets composed of immiscible liquids entrained in threads, with control of mass transfer processes to produce core fibers/shells with desired morphologies.

Wu et al. [50] reported a novel carbon fiber/epoxy multiscale hybrid composite reinforced with core-shell nanofibers. The ultrafine fibers were manufactured by co-electrospinning, in which liquid dicyclopentadiene (DCPD) was wrapped in polyacrylonitrile (PAN) to form DCPD/PAN nanofibers. The experimental results indicate that the flexural stiffness of such a new composite after failure can be completely recovered by the intermediate layers of nanofiber, acting as self-healing agents. SEM micrographs evidenced the release of curing agents at laminated interfaces and the hardening and self-healing mechanisms of core-shell nanofibers.

These fibers present great innovative potential and are in development for diverse applications. However, some essential studies are still needed, and some challenges still need to be addressed. Obtaining such fibers with the desired properties on a large scale is an important issue that must be considered. Mechanical behavior such as tensile strength plays a key role in determining yarn production capacity. There are still some unknown and uncontrollable parameters that need more attention, and studies should be carried out on such factors that influence the characteristics of this type of nanofiber [33].

#### 2.2.3. Electrospinning with Alternating Corona Current

Alternating current (AC) electrospinning can achieve high nanofiber generation rates while adding more flexibility to process development when compared to direct current (DC) electrospinning. However, the AC electrospinning process can produce very different results than DC electrospinning when using the same precursors [51]. Research in electrospinning from different areas generally used high-voltage sources of direct current (DC) to generate the electrostatic field [30,52,53]. The simplest attempt to scale up was the introduction of multiple spinnerets, although it proved to be a challenge due to the constant clogging of the spinning ends [54]. Other approaches been under trial of development, such as co- and triaxial electrospinning [55,56] and needleless free surface methods [57]. Despite this, replacing the high voltage of static direct current with a high voltage of dynamic alternating current can lead to significant differences in fiber formation [58].

Electrospinning with alternating corona current has only been reported for a few cases [59,60,61] in contrast to the vast literature on direct current [62,63]. Paulett et al. [51] demonstrated that alternating current electrospinning produced uniform and mechanically strong polyacrylonitrile nanofibrous (PAN) meshes at voltages of 30 ± 5 kV RMS when 0.75–6.0 wt.% nanocrystalline cellulose-II was added in a typical PAN precursor solution. Efficient generation (rate of up to 2 g/h or mass flow of 0.7 g/h.cm²) of nanofibers with diameters of 250–500 nm has been observed when using flat fiber generation electrodes with diameters of up to 25 mm.

Farkas et al. [58], aiming at this increase in productivity, used the alternating current corona electrospinning process combining the intense forces of the alternating electrostatic field and a free surface needle spinner design. Productivity reached two orders of magnitude higher (up to 1200 mL/h) than classical single needle direct current electrospinning without any change in fiber properties.

Even if it is simple to make this change, by replacing the high-voltage direct current generator with an alternating current power supply for example, the effects on morphologies and whether the increase in productivity compensates for possible losses in fiber quality are still under investigation. Furthermore, the use of alternating current with rotating drums, which expects even higher transfer rates compared to high direct DC voltage, does not present comprehensive studies [64,65].

#### 2.2.4. Electret Fibers

The growing demand for air filters with high filtration efficiency and low-pressure drops led several authors to seek ways to achieve this objective in an economically viable way. Fibrous air filters allow capturing air particles through the synergistic effect of thick physical barriers and adhesion [66]. However, one strategy to increase efficiency is to increase fiber consumption, increasing the size and thickness of the filter and its protection capacity. However, at the same time, this strategy causes a sharp increase in the pressure drop, which can be a disadvantage in certain applications [67].

One of the ways to alleviate this conflict is the use of electret fibers, which are obtained through polarized dielectric materials, and which are electrified by processes such as corona charging, triboelectric charging, and induction charging [67,68]. Typically, the definition of an electret is a permanently polarized and electrically insulating material with a quasi-permanent and long-lasting internal or external surface charge [69].

Due to the use of high voltage during electrospinning, various polymers such as polyvinylidene fluoride [70,71,72], polyetherimide [73], polyacrylonitrile [74], and polyamide [75] can store electrical charges in spun fibers. The effect of the amount of charge held by a filter on its filtering efficiency is quite clear: the greater the charge, the greater the electric field produced and the greater the efficiency of collecting particles by electrical forces [76]. However, the useful life of these fillers and their possible neutralization by capturing oppositely charged particles make a barrier to the use of these fibers. When the charge is reduced or neutralized, the dominant capture mechanisms become direct interception and Brownian motion, and the electret fiber is reduced in performance to an uncharged material of similar fiber size and structure. In addition, the weak electrostatic force produced by electret membranes subjected to electrospinning may limit their performance as filter media [77].

Based on this, several studies have focused on the evaluation of different polymers and ways to add loads to the fibers in order to increase the useful life of the loads and ensure high-efficiency operation. One of the ways to accomplish this is the introduction of nanoparticles as an enhancer for charge storage, as illustrated in Figure 6a. Along these lines, Huang et al. [77] used electrospinning to manufacture polyvinylidene fluoride (PVDF) electret membranes with a new charge storage enhancer through nanoscale graphite platelets (NGP) dispersed within the fibrous matrix. It has been shown that methodologies to improve the disposition of NGP inside the fibers are an impasse to improve the surface potential of membranes. It has been shown that there is a limit to the NGP load required for optimal filtration efficiency and pressure drop. The highest quality factor was obtained for PVDF fibers filled with 2% by weight NGP under a high face speed of 1.26 m/s. The composite membrane demonstrated a high filtration efficiency of 98.989%, with a low pressure drop of 1279 Pa and a quality index of 3.591 kPa^−1^.

Lolla et al. [70] also used polyvinylidene fluoride (PVDF) for membrane electrospinning under controlled conditions. Fiber polarization was performed using a custom-made device of a Teflon^®^ structure and aluminum electrodes to enhance the formation of the β-phase ferroelectric property of the fiber electret by simultaneous uniaxial stretching of the fiber mat and heating the mat up to Curie temperature (150 °C, 20 min) of the PVDF polymer in a strong field of 2.5 kV/cm. MATLAB simulations revealed changes in electric field paths and magnetic flux within the polarization field with the inclusion of ferroelectric fibers. The electrostatic particle capture mechanism was dominant, as indicated by SEM images, which showed captured particles distributed as many individual particles attached to many fibers. In comparison, particles captured by unpolarized fibers tended to form internal and surface particle cakes. Strong electrostatic forces contributed to the high collection efficiencies of these media.

Song et al. [78] used another polymer, polyacrylonitrile (PAN), and Fe_3_O_4_ magnetic particles. These particles were in the form of polyhedral oligomeric silsesquioxanes (POSS) with Si–OH and were prepared by hydrosilylation reaction between Fe_3_O_4_-SiH and POSS with hydroxyl and vinyl groups. Electret nanofiber properties showed remarkably better surface potential stability, with surface potential retention of up to 50% for PAN with 1% by weight Fe_3_O_4_-POSS. Compared to pure PAN, charge retention has been increased by 21%. Following the line of use of polyacrylonitrile, and to increase its strength and mechanical properties, Gao et al. [74] proposed performing free surface electrospinning with multiple jets using scaffold nanofibers, microspheres, and thin nanofibers as ternary structures. The scaffold nanofibers constituted a stable skeletal structure in which the microspheres were embedded, these microspheres increased the interfiber voids, thus reducing the pressure drop, while thin nanofibers with a diameter of 84 nm intertwined with scaffold nanofibers improved the probability of collision of airborne particles and ensure robust filtration performance without sacrificing filtration efficiency.

Following a new line, Li et al. [67] prepared fully polymeric hybrid electret fibers (without the inclusion of charge-storing nanoparticles), studying the complementarity of electrical responses between electrospun polymers. Figure 6b represents how the charges are on the surface of the nanofiber. The coupling of electrical polarization behaviors of polystyrene (PS), with a low dielectric constant, and polyvinylidene fluoride (PVDF), with a high dielectric constant, allowed the hybrid PS/PVDF fibers to exhibit an enhanced electret effect and high porosity. Charges injected into the matrix of PS fibers generated local electric fields exhibiting the same direction as the external electrostatic field; however, the orientation of the PVDF dipoles generated local electric fields in the opposite direction to the external one. An interesting point of the work is that the electret property was improved without relying on toxic nanoparticles as charge enhancers, which was never reported in previously developed electrospun electret fibers and melt-cast electret fibers.

In another study, Ignatova et al. [79] tried to use the corona discharge method to store charges in nanofibrous electrets of poly(ethylene terephthalate) prepared by electrospinning. The system consisted of a corona electrode (connected to a high voltage source), a grounded plate electrode, and a grid placed between them. In preliminary experiments, loading parameters such as stresses, time, and distances were optimized. The storage of charge in the specific surface area, through surface potential studies, showed that the later application of corona charge produces differences in the initial time stability, which is valid for both types of corona polarity. However, the differences in charge obtained from peak assessments in TSC measurements near the glass transition temperature of corona-charged PET are largely due to the relaxation within the PET fibers by the charges deposited on the sample surface during the corona discharge process.

Research in this area is still fragmented and not complete enough to draw precise conclusions. Therefore, an overview is considered necessary when evaluating electret air filters. This review summarizes the research progress made so far on electret air filters and provides the need for specific research in this area to further improve air filtration performance and the long-term effectiveness of electret air filters [68].

#### 2.2.5. Addition of Nanoparticles In Situ

Nanoparticles are nanometer-scale particles (10^−9^ m) that appear in different forms in the environment. Particles of this size have unique characteristics about macroscopic particles and surfaces [1]. ISO/TS 12025:2012 defines these particles as those that comprise the nanometer scale, between 1 and 100 nm. The unique and novel characteristics that ultrafine particles and nanoparticles present are associated with their extremely small size, chemical composition (purity, crystallinity, electronic properties), surface structure (surface reactivity, surface groups), solubility, shape, and aggregation [80,81]. These different properties, together with their small size and surface area, have motived the study of nanometer-scale particles and their application in several areas. This was accentuated in universities, industries, and emerging companies in this field. These advances in the areas of nanoscience and nanotechnology have resulted in numerous possibilities for applications in consumer products [4,82,83].

Regarding electrospinning, the addition of nanoparticles to fibers is being studied to confer differentiated properties and increase the applicability of these materials in the field of advanced materials science, due to their remarkably improved thermal, chemical and dimensional stabilities, applicability, electrical conductivity, mechanical, and functional [84].

In general, the pre-electrospinning or in situ modification technique is considered the simplest method of membrane modification. In pre-treatment, additives are added directly to the spinning solution, as schematized in Figure 7. The establishment of functionalizing agents in the spinning solution presents new formulations with specific properties. It is considered the simplest method of modifying membrane performance [85]. One of the problems with this process is that the nanoparticles are covered with nanofibers, as can be seen in the cut of the electrospun fiber (Figure 7).

Khalili and Chenari [86] performed the electrospinning fabrication of PVP/zirconia nanofibers followed by different heat treatments. SEM images show the formation of smooth, granule-free fibers with a mean diameter below 100 nm after calcination. The room temperature magnetization results showed a hysteresis loop formation, indicating the ferromagnetic behavior of the samples. In the same vein, Sundarrajan et al. [87] manufactured a new PVA membrane system with the introduction of nickel/zirconium oxide as additives. Likewise, Annur et al. [88] reported an easy method for developing chitosan antibacterial membranes through the immobilization of silver nanoparticles. The incorporation of silver nanoparticles into the chitosan spinning solution showed a considerable increase in antibacterial properties. Bortolassi et al. [89] evaluated the addition of silver (Ag) to nanofibers of polyacrylonitrile (PAN) to be used as air filters. Ag nanoparticles gave the filters antibacterial activity. 

### 2.3. Fiber Modifications after Electrospinning Process

#### 2.3.1. Nanoparticles Addition Techniques

The functionality of nanofibers is directly linked to the nanoparticle impregnated during or after the electrospinning process. Depending on the desired applicability of the nanofibers/nanoparticles, nanoparticles of metal oxides, zeolites, enzymes, adsorbents are used together with the nanofibers [90]. Different physical/chemical or biological properties can be obtained [91].

The modification of nanofibers using techniques after electrospinning is a promising alternative to promote surface changes. It provides desired characteristics by introducing new chemical groups on the surface of nanofibers, converting already present functionalities into more desirable ones, or removing portions of the existing surface. In addition, it is economically viable on an industrial scale [85,92]. A wide variety of materials and techniques used to carry out modifications for different purposes can be used to disperse nanoparticles in polymeric fibers [85,93,94].

#### 2.3.2. Reinforcing Composite Materials

The nanofiber can be used together with composite materials, to improve the mechanical proprieties of composites, such as the interlaminar fracture toughness and damping [95]. Among the advantages of adding fibers to composites is their favorable specific stiffness, specific strength, and ability to dissipate energy [96]. Daelemans et al. [97] related coaxial PA6/PCL nanofibres interleaved composite laminates made using an in-house developed vacuum-assisted resin transfer molding. The composite produced did not need additional chemical modification steps or specialty polymers. Improved adhesion of nanofibers occurred with the use of core-shell structured nanofibers with an interdiffusion shell.

When looking for high mechanical performance and high lightness, carbon-fiber-reinforced polymers (CFRPs) are used. However, they suffer from delamination and low damping. Thus, the use of intercalated nanofibers is an alternative to improve these properties, as reported by Maccaferri et al. [37] and Zheng et al. [98]. Maccaferri et al. [37] produced nitrile butadiene rubber/poly(ε-caprolactone) (NBR/PCL) blend rubbery nanofibrous integrated into epoxy CFRP laminates. The results proved the improvement of the damping and interlaminar fracture toughness of CFRP with the intercalation of the nanofibers. The authors also showed that the choice of nanofiber weight, the number, and the position of intercalations affected the damping of CFRP. Zheng et al. [98] prepared nylon (PA66)/polycaprolactone(PCL) blend nanofiber interleaved carbon fiber/epoxy (CF/EP) by the lay-up procedure, under vacuum, and cured in a hot press. The increased interfacial adhesion between the PCL coated PA66 nanofiber and the epoxy matrix contributed to a greater improvement in fracture energy.

The technique presented allows engineers to design through the addition of ultra-low weight rubberized nanofibrous layers advanced composite components with high damping capacity. Thus, the nanomodified composite material is suitable for applications that require high energy dissipation while ensuring high mechanical performance [96].

#### 2.3.3. Surface Treatment

One of the most widely studied electrospinning post-treatments when working with nanoparticles is the surface treatment of electrospun nanofibers. Figure 8 shows one of the surface treatment techniques, the coating, which consists of immersing electrospun fibers directly in a solution of nanoparticles. Electrostatic forces, hydrogen bonds, or interactions between functional groups are formed between electrospun fibers and nanoparticles. One of the problems with this technique is that nanoparticles generally tend to aggregate, forming defects in the fiber structure and loss of applicability, as represented in Figure 8 [93]. Another technique for surface treatment of electrospun nanofibers is the spraying of nanoparticles on the electrospun nanofibers (electrospray-coating), also represented in Figure 8. The challenge of this technique is the heterogeneous dispersion of the nanoparticle on the polymeric surface, avoiding the formation of nanoparticle aggregates [99].

Tajer, Anbia, and Salehi [100] evaluated polyacrylonitrile (PAN) nanofibers containing micro and nanoparticles of activated carbon for adsorption of toxic gases (SO_2_, CO_2_, and CH_4_). Activated carbon particles were deposited on PAN nanofibers by the electrospray technique. Both particles were effectively dispersed on the surface of the nanoparticle, making the compound an efficient adsorbent, mainly for SO_2_.

Selatile et al. [99] compared coating and electrospray-coating techniques on the effectiveness of polylactic acid (PLA) nanofibers in making Ag nanoparticles available for antimicrobial activity. The authors showed that the dispersion was effective in both techniques, showing good microbial activity. The PLA/Ag nanofibers obtained by electrospray-coating showed higher microbial activity than those obtained by coating, making them promising for applications such as air filter membranes. Razzaz et al. [101] studied two methods of impregnation of titanium oxide (TiO_2_) nanoparticles: chitosan nanofibers coated with TiO_2_ nanoparticles by the coating technique; and electrospinning of chitosan/TiO_2_ solutions. They evaluated the adsorbent potential of nanofibers for the removal of heavy metals (Pb (II) and Cu (II) ions). They concluded that the chitosan/TiO_2_ nanofibers obtained by direct electrospinning of the solution of both compounds showed a greater potential for metal absorption compared to the chitosan fibers coated with TiO_2_, a fact that can be attributed to the easy diffusion of metals in the pores of the chitosan/TiO_2_ compound. Al-Dhahebi, Gopinath, and Saheed [102] analyzed the use of post-treatment techniques for electrospun fibers with graphene, such as physical immersion coating, ultrasound, plasma treatment, wet chemical method, and radiation treatment. They showed that post-processing methods usually have greater efficiency in the use of graphene particles for biosensor applications since it presents a large surface area of nanofibers prepared with graphene in a heterogeneous way across the surface of the nanofiber. The disadvantage of post-processing techniques lies in their ability to establish precise interactions between nanoparticles and polymer nanofibers. Therefore, studies are being developed to optimize impregnation techniques or generate new deposition strategies on the surface of electrospun nanofibers [102,103].

One of the methods studied to improve particle/nanofiber adhesion is plasma treatment. The method consists of modifying the surface of the material without changing its volume characteristics, using plasma (partially or fully ionized gas formed under specific conditions and composed of ions, electrons, and free radicals, which can potentially interact with contact materials) [104]. Among the gases commonly used for surface, treatments are nitrogen, ammonia, oxygen, hydrogen, argon, helium, and carbon dioxide, which makes the technique highly versatile. Plasma treatment is mainly used to adjust surface adhesion and roughness [85]. In addition, it allows the increase in efficiency of physical absorption in hydrophobic nanofibers, creating a more hydrophilic surface, thus increasing the fixation of biomolecules [102]. Another advantage of plasma treatment is that it is environmentally friendly (avoiding the use of solvents) [104].

Shaulsky et al. [105] coated poly(vinylidene fluoride-hexafluoropropylene fluoride) (PVDF-HFP) electrospun nanofibers with polyvinylidene fluoride (PVDF) through phase separation. The purpose of such coating was to adjust the pore size, porosity, and morphology of the fibers. In the phase separation process, the solid polymer structure collapses into the voids of the substrate. Thus, the polymer-coated nanofibers reduced the voids between the fibers. Another contact was the 20.4% decrease in the porosity of the coated nanofibers. The authors concluded that the proposed electrospun nanofiber coating technique may be suitable for air and water filtration applications based on the found pore size.

The use of heat treatments in electrospun nanofibers can be used mainly to improve the pore size distribution and the mechanical properties of the fibers. Electrospun fibers are randomly distributed, which results in lower structural integrity and low mechanical strength. Heat treatment can improve the mechanical properties and structural integrity of fibrous structures. Therefore, heat treatment can be considered a promising method to increase the mechanical strength of electrospun membranes [85]. Another method used to improve the mechanical properties of electrospun nanofibers is the addition of nanoparticles, such as graphene oxide [106].

Nauman, Lunineau, and Alharbi [107] presented a review of the post-treatment methods used to improve the mechanical properties of electrospun nanofibers. The authors addressed the techniques of polymeric crosslinking, fiber stretching, solvent welding, heat treatment, pressing, and hot stretching. They summarized the effects of each treatment on the main mechanical properties and showed that all treatments increased tensile strength and Young’s modulus, directly related to fiber stiffness.

## 3. Final Considerations

Fibers produced by electrospinning have properties such as high surface area, high porosity and nanoscale dimensions that make them attractive for several applications in different areas. The modification of these fibers with treatments during or after electrospinning makes them even more specific and functionalizes them for each field of application. Therefore, this review considered a macro view on different strategies to develop fibers as of the electrospinning technique, functionalizing them with different types of treatments or incorporation of materials. Thus, depending on the type of final fiber application, the process variables are determined. These variables range from the polymer used and the electrospinning conditions to the treatment and addition of nanoparticles. There are still many challenges regarding the modification of fibers produced by electrospinning. Several factors change the properties of the fibers and the combination between them is important for the specificity of the fiber to be produced. The search for new and increasingly specific materials continues to this day. Due to this, new techniques have been developed, as well as combinations of different materials in the same component. In the area of air filtration, one of the great challenges is the production of nanofibers on a large scale that combines mechanical, biological, and chemical properties in the same material.

## Figures and Tables

**Figure 1 membranes-12-00861-f001:**
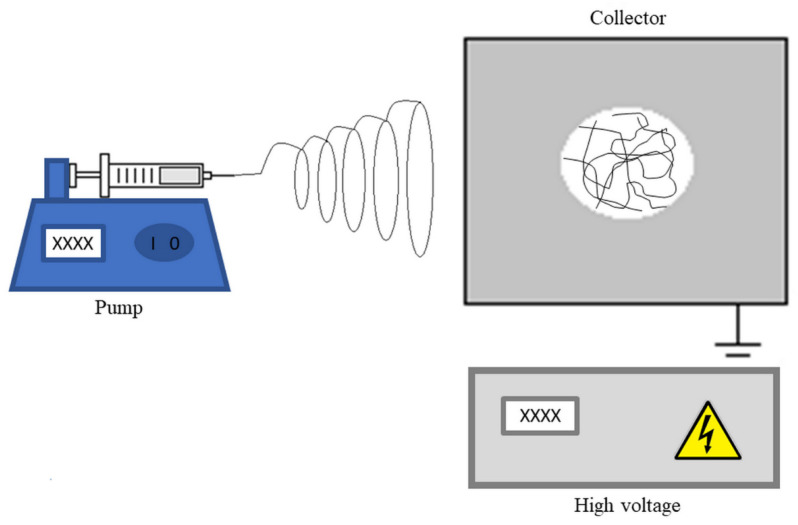
Electrospinning scheme.

**Figure 2 membranes-12-00861-f002:**
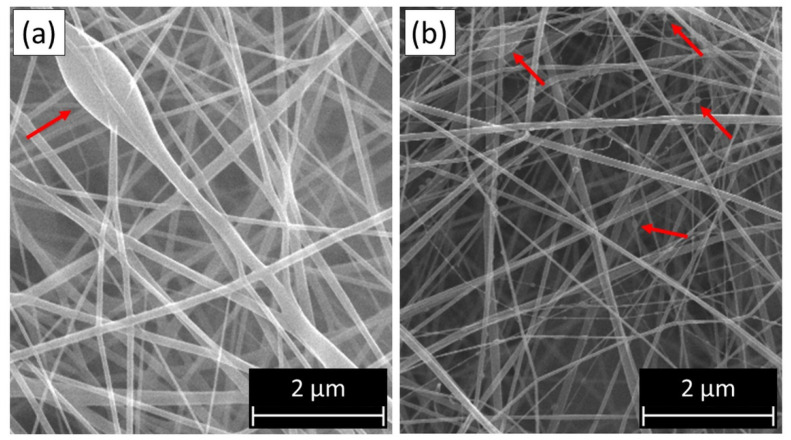
Formation of beads on electrospun fibers: (**a**) large beads; (**b**) small beads.

**Figure 3 membranes-12-00861-f003:**
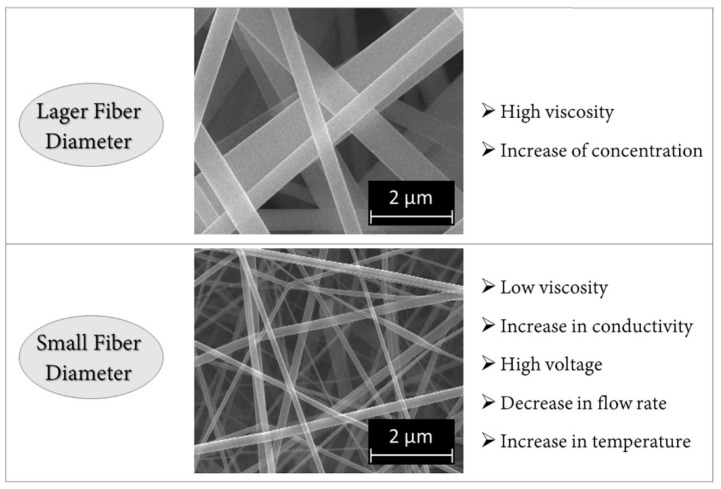
Summary of electrospinning parameters and their effects on fiber diameter.

**Figure 4 membranes-12-00861-f004:**
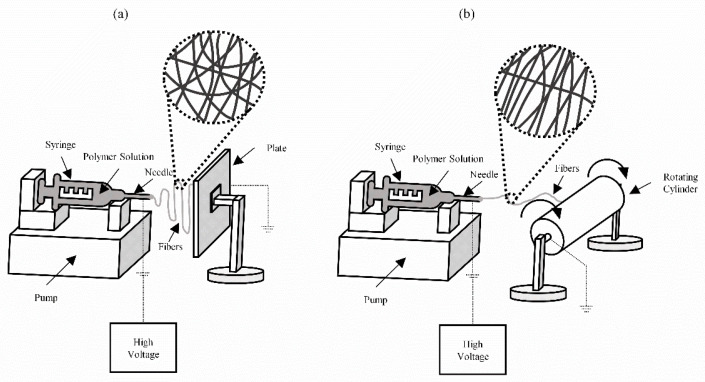
Electrospinning setups and collector types: (**a**) flat plate; (**b**) drum collectors.

**Figure 5 membranes-12-00861-f005:**
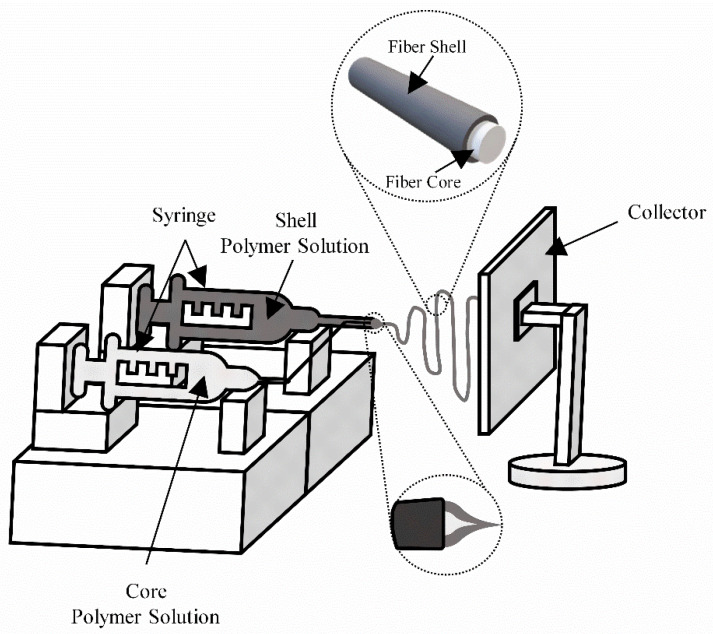
Core/shell electrospinning technique schema.

**Figure 6 membranes-12-00861-f006:**
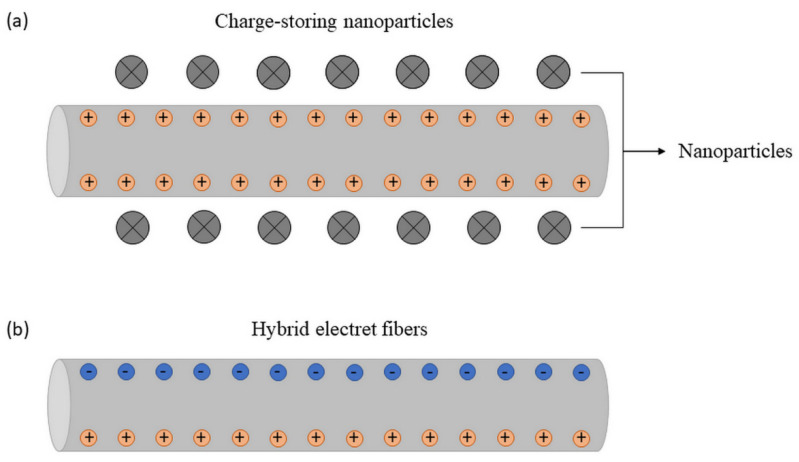
Electret fibers: (**a**) nanoparticles as an enhancer for charge storage; (**b**) fully polymeric hybrid.

**Figure 7 membranes-12-00861-f007:**
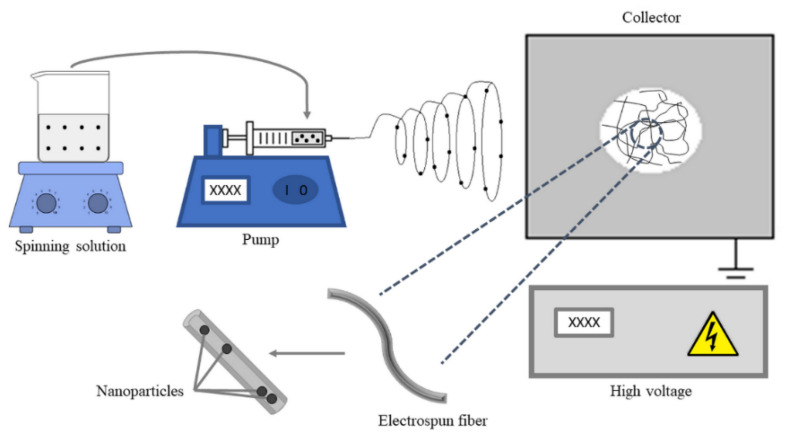
In situ modification technique schema.

**Figure 8 membranes-12-00861-f008:**
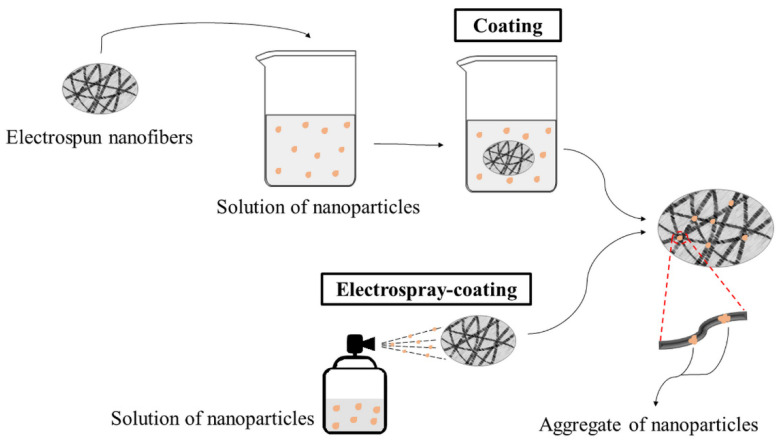
Coating and electrospray-coating modifications techniques schema.

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
