# Peer review of "Modification and Functionalization of Fibers Formed by Electrospinning: A Review"

_membranes, 2022, doi:10.3390/membranes12090861_

Round 1

Reviewer 1 Report

Cnts for Authors are in the attached pdf file.omme

Author Response

Dear Editor,

Firstly, we would like to thank the editor and reviewers for their contributions to improving the quality of our manuscript by posing important questions and making useful comments. All questions and comments of the reviewers were taken into account to elaborate the revised version of the manuscript titled “Modification and Functionalization of Fibers Formed by Electrospinning: A Review”, which we are resubmitting to be considered for publication in Membranes. Also attached is a response sheet with a point-by-point list of answers to the reviewer´s questions and comments.

Best regards,

Reviewer 2 Report

The proposed review contains some information regarding a specific application field of nanotechnology, more specific, regarding the fabrication and applications electrospun polymer nanofibers. Indicating many facts, the authors give no explanations of observed regularities. Such review type is wide used last time, so if such a review type is suitable for editor, the proposed manuscript can be published.

Indicating many facts, the authors give no explanations of observed regularities. Such review type is wide accepted last time, so if such a paper type is suitable for editor, the proposed manuscript can be published.

Note that there are few inaccurate statements in the manuscript beginning, regarding some basic points of theory of electrospinning process (perhaps such a situation due to fact that the authors are more related in their research with electrospun nanofiber applications, but not in theoretical aspects of electrospinning process).

So, the below inaccurate statements should be corrected.

Page 3

Paragraph 1:

First of all, the diameters and quality of electrospun fibers are affected by polymer solution concentration.

Paragraph 2:

The problem in spinability at low concentration is related to elastic solution properties, not to its viscosity.

Paragraph 3:

The surface tension of the solution is important only for kinetic of Taylor cone formation and spinning process starting, thereafter it is of no importance.

Page 6, line 168

In place of “flexural instabilities it is better to use “bending instability”, as was formulated in the firsts papers, devoted to this phenomenon.

After the necessary modification the manuscript can be recommended for publication.

Author Response

(The authors gave the same response as above.)
